# DM-Tune: Quantizing Diffusion Models with Mixture-of-Gaussian Guided Noise Tuning

## Abstract

Diffusion models have become essential generative tools for tasks such as image generation, video creation, and inpainting, but their high computational and memory demands pose challenges for efficient deployment. Contrary to the traditional belief that full-precision computation ensures optimal image quality, we demonstrate that a fine-grained mixed-precision strategy can surpass full-precision models in terms of image quality, diversity, and text-to-image alignment. However, directly implementing such strategies can lead to increased complexity and reduced runtime performance due to the overheads of managing multiple precision formats and casting operations. To address this, we introduce *DM-Tune*, which replaces complex mixed-precision quantization with a unified low-precision format, supplemented by noise-tuning, to improve both image generation quality and runtime efficiency. The proposed noise-tuning mechanism is a type of fine-tuning that reconstructs the mixed-precision output by learning adjustable noise through a parameterized nonlinear function consisting of Gaussian and linear components. Key steps in our framework include identifying sensitive layers for quantization, modeling quantization noise, and optimizing runtime with custom low-precision GPU kernels that support efficient noise-tuning. Experimental results across various diffusion models and datasets demonstrate that DM-Tune not only significantly improves runtime but also enhances diversity, quality, and text-to-image alignment compared to FP32, FP8, and state-of-the-art mixed-precision methods. Our approach is broadly applicable and lays a solid foundation for simplifying complex mixed-precision strategies at minimal cost.

## 1 Introduction

Diffusion models serve as potent tools for tasks like image generation, video creation, and inpainting, yet they demand significant computational and memory resources Ramesh et al. (2021); Saharia et al. (2022); Rombach et al. (2022b); Liu et al. (2024). For instance, generating a single image using Stable Diffusion XL (SDXL) Podell et al. (2023), which has approximately 10 billion parameters, requires over two minutes on an A100 GPU, highlighting the urgency for efficiency enhancements. Recent AI hardware advancements, particularly GPUs, often rely on low-precision computation to improve performance. For instance, Nvidia's next-generation GPUs, like the Blackwell chips, support a variety of floating-point (FP) formats (from FP64, FP32, TF32, FP16, and BF16, down to FP8, FP6, and FP4), yielding a vast search space for efficient execution of deep learning models.

Quantization is an effective approach to leverage the low-precision capability of the hardware, which can reduce memory and computational costs. However, quantizing the whole model and activations to a low-precision format (e.g., 8-bit) often leads to low-quality images in diffusion models. To alleviate this problem, previous research has explored mixed-precision quantization with complex strategies (*i.e.*, intra-layer, timestep-aware) Li et al. (2023a); Shang et al. (2023); He et al. (2024b). However, mixed-precision quantization for diffusion models in the current literature suffers from two main problems. *1) Full-precision fallacy:* The first is that there is traditional wisdom that full-precision computation provides the highest accuracy. However, our research has revealed a proper FP-based mixed-precision (FP-MP) quantization strategy can *outperform full-precision* image generation quality and diversity, provide more details, and better follow the input prompt. This is shown in Fig. 1 (a), which compares the generated images in FP32 and FP-MP (mixture of BF16 and FP8) for a Stable Diffusion model. This is because diffusion models (unlike other deep learning methods) deal with probabilistic distributions with random data and the introduced quantization noise is

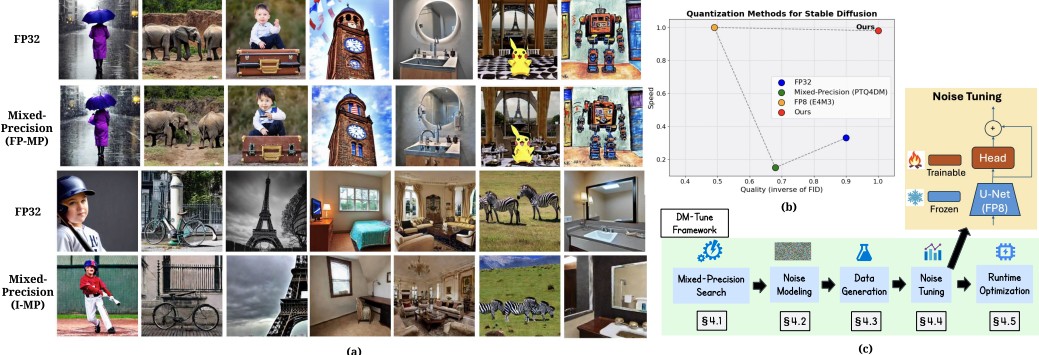

Figure 1: (a) Generated images in FP32, FP-based mixed-precision (mixed BF16 and FP8), and integer-based mixed precision (Q-Diffusion) using a fixed seed. FP-MP offers greater detail and better prompt alignment compared to FP32. For example, in the prompt "*a view of a multi-tiered clock tower with a US flag on the top*," FP32 misplaces the flags, leaving them suspended in the sky, whereas FP-MP correctly positions a flag at the top of the tower. In contrast, I-MP struggles with image regeneration, particularly when generating human faces. (b) Comparing image quality and speed of our approach with PTQ4DM, FP32, and FP8. (c) Overview of DM-Tune.

absorbed into the inherent noise of the diffusion process, potentially improving the metrics. Prior art focused on integer-based mixed-precision quantization. However, we believe that the nonlinear quantization noise from FP (rather than INT) plays a vital role in performance advantage. We compare a prior art PTQ4DM Shang et al. (2023) that uses integer-based mixed-precision (I-MP) with full-precision in Figure 1 (a). I-MP not only fails to improve image generation quality and prompt alignment but also produces low-quality images. *2) Mixed-precision overhead:* The second challenge is that mixed-precision quantization can be slower even compared to full-precision due to the complicated quantization strategies and casting overheads. Additionally, low-precision resources that are critical for achieving the peak throughput are not fully exploited.

To solve these problems, we propose *DM-Tune*, a framework that first identifies an efficient FP-MP strategy that **outperforms** full-precision by progressively introducing *controlled noise* to the model. Then, it replaces the FP-MP model with a **unified low-precision** model, coupled with a **noise-tuning** head, to optimize both image generation quality and speed. Noise-tuning is a form of fine-tuning that adds a guided noise head, learned through trainable parameters, to the model's low-precision output to recover the FP-MP output. Noise-tuning has low overhead, as only the added parameters for the head are trained and applied once before inference.

To achieve this goal, three challenges need to be addressed. First, the search space for exploring the optimal FP-MP strategy to outperform full-precision is vast. The second challenge is how the operator of noise-tuning needs to be expressed. Third, deploying the noise-tuning during the inference slows down the diffusion model since the noise-tuning operator is memory-bound. To resolve the first challenge, we first identify sensitive layers and then propose two novel techniques to enhance the quality: *prompt-aware* and *timestep-aware quantization*. Second, we recognize that multiple overlapping Gaussians are needed as nonlinear functions to recover FP-MP output. Finally, we provide a highly optimized GPU kernel that fuses matrix multiplication in low-precision with nonlinear Gaussian terms. Our approach significantly reduces the runtime of diffusion models compared to state-of-the-art (SOTA) methods, while also enhancing image quality compared to both SOTA and full-precision models (Figure 1 (b)). Figure 1 (c) shows the overview of this work. This paper makes the following contributions:

- We show that a novel mixed-precision strategy can outperform full-precision image generation quality and diversity for different types of diffusion models.

- We propose a new technique called noise-tuning, which enables running the model in low-precision at high speed while achieving mixed-precision quality in a data-free manner.

- We develop highly optimized GPU kernels that fuse matrix multiplication in low-precision with nonlinear functions.

- Experimental results show that our approach improves the runtime of prior art by $5.2\times$ while improving image generation quality and diversity.

## 2 BACKGROUND

**Diffusion Models:** Diffusion models generate images using a Markov chain process. Initially, a forward diffusion process adds Gaussian noise to the data $x_0 \sim q(\mathbf{x})$ over $T$ steps, resulting in progressively noisier samples $\mathbf{x}_1, \ldots, \mathbf{x}_T$:

$$q(\mathbf{x}_t|\mathbf{x}_{t-1}) = \mathcal{N}(\mathbf{x}_t; \sqrt{1-\beta_t}\mathbf{x}_{t-1}, \beta_t\mathbf{I}) \tag{1}$$

Here, $\beta_t \in (0, 1)$ is a variance schedule that determines the intensity of Gaussian noise at each step. As $T \to \infty$, $\mathbf{x}_T$ converges to an isotropic Gaussian distribution.

The backward process removes noise from a sample drawn from the Gaussian noise input $\mathbf{x}_T \sim \mathcal{N}(0, \mathbf{I})$ to generate high-fidelity images. Since the actual reverse conditional distribution $q(\mathbf{x}_{t-1}|\mathbf{x}_t)$ is unknown, diffusion models sample from a learned conditional distribution:

$$p_\theta(\mathbf{x}_{t-1}|\mathbf{x}_t) = \mathcal{N}(\mathbf{x}_{t-1}; \tilde{\mu}_\theta(\mathbf{x}_t), \Sigma_\theta(\mathbf{x}_t, t)) \tag{2}$$

$$\tilde{\mu}_\theta(\mathbf{x}_t) = \frac{1}{\sqrt{\alpha_t}}\left(\mathbf{x}_t - \frac{\beta_t}{\sqrt{1-\bar{\alpha}_t}}\epsilon_\theta(\mathbf{x}_t, t)\right) \tag{3}$$

where $\alpha_t = 1 - \beta_t$ and $\bar{\alpha}_t = \prod_{i=1}^{t}\alpha_i$. The variance $\Sigma_\theta(\mathbf{x}_t, t)$ can either be reparameterized or set to a constant schedule $\sigma_t$. When using a constant schedule, $\mathbf{x}_{t-1}$ is given by:

$$\mathbf{x}_{t-1} = \frac{1}{\sqrt{\alpha_t}}\left(\mathbf{x}_t - \frac{\beta_t}{\sqrt{1-\bar{\alpha}_t}}\epsilon_\theta(\mathbf{x}_t, t)\right) + \sigma_t\mathbf{z} \tag{4}$$

where $\epsilon_\theta(\mathbf{x}_t, t)$ is the noise estimation model output at timestep $t$. The U-Net architecture Ronneberger et al. (2015) is predominantly used in designing the noise estimation model. For further details, we refer readers to Ho et al. (2020). This work focuses on quantization of the U-Net during the inference (backward process).

**Evaluation of Diffusion Models:** The evaluation of diffusion models is different from other deep learning models. First, the evaluation can be unfair and biased. Second, only one metric (i.e., accuracy in image classification) is not sufficient to evaluate the model as the distribution of generated images can suffer from different phenomena. These phenomena include memorization, mode collapse, mode shrinkage, mode invention, and density shift Alaa et al. (2022). To address these challenges, we adopt a similar approach to Stein et al. (2023) by using *DINOv2-ViT* model as the feature extractor instead of the traditional *Inception-V3*. Also, we evaluate the models comprehensively with different metrics. We use Fréchet Inception Distance (FID), Kernel Distance (KD), and Sliced-Wasserstein Distance (SW) to evaluate overall image generation performance Stein et al. (2023). Additionally, we assess CLIP (for prompt alignment), precision and density (for image quality), recall and coverage (for diversity), and authenticity (to ensure the model does not replicate images from the training data) metrics Naeem et al. (2020). For FID, KD, and SW, the lower is better but for other metrics, the higher is better.

## 3 MOTIVATION

We summarize the key motivations of this work.

**1) Mixed-precision outperforms full-precision:** Contrary to the traditional wisdom that assumes full-precision calculation provides the highest image generation quality, FP-MP has the potential to outperform its full-precision counterpart in diffusion models. This is due to two main reasons. First, the diffusion model is a stochastic process (i.e., the seed for the initial state), meaning that certain seeds may yield better image generation results. Second, the quantization noise is absorbed into the noise introduced by the scheduler, which can alter the diffusion trajectory in a beneficial direction. To understand this, we have the following equation if we rewrite the Eq. 4 for the quantized model:

$$\begin{aligned}
\mathbf{x}'_{t-1} &= \frac{1}{\sqrt{\alpha_t}}\left(\mathbf{x}'_t - \frac{\beta_t}{\sqrt{1-\bar{\alpha}_t}}\left(\epsilon_\theta(\mathbf{x}'_t, t) + \Delta\epsilon_\theta(\mathbf{x}'_t, t)\right)\right) + \sigma_t\mathbf{z} \\
&= \frac{1}{\sqrt{\alpha_t}}\left(\mathbf{x}'_t - \frac{\beta_t}{\sqrt{1-\bar{\alpha}_t}}\epsilon_\theta(\mathbf{x}'_t, t)\right) - \frac{\beta_t}{\sqrt{\alpha_t(1-\bar{\alpha}_t)}}\Delta\epsilon_\theta(\mathbf{x}'_t, t) + \sigma_t\mathbf{z}
\end{aligned} \tag{5}$$

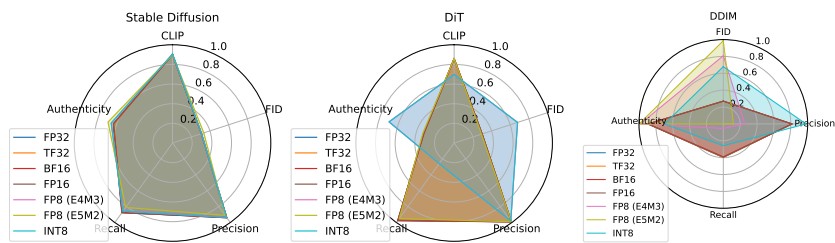

Figure 2: Evaluating three diffusion models for different data formats.

where $\mathbf{x}'_t$ and $\Delta\epsilon_\theta(\mathbf{x}'_t, t)$ are the quantized input and U-Net output quantization error at timestep $t$, respectively. We assume that the benefit of mixed-precision quantization is unique to floating-point formats (FP-MP), as we observe that I-MP does not offer significant improvements in image generation. This is because integer quantization has a limited dynamic range and the quantized numbers are clipped to the maximum/minimum. Therefore, a complex mixed-precision strategy is required to maintain high-quality image generation with integer-based quantization.

**2) Mixed-precision slows down:** The mixed-precision designs (both I-MP and FP-MP) suffer from poor speed due to the overhead of complicated quantization strategies (it can be even slower than full-precision). Recent AI hardware advancements, particularly GPUs, often rely on lower precision computation to enhance performance. To optimize speed, we found that using unified low-precision for the entire U-Net while adding a nonlinear function with a negligible number of parameters (compared to the total parameters) is enough to recover FP-MP output. Our proposed FP-MP will be introduced at the end of Sec. 4.1.

## 4 DM-TUNE

In this section, we describe the components of DM-Tune framework. These are: *mixed-precision search*, *noise modeling*, *data generation*, *noise-tuning*, and *runtime optimization*. Our methodology provides a generalized and scalable solution that can be adapted to different models, datasets, and quantization techniques.

### 4.1 MIXED-PRECISION SEARCH

In this step, we first select two precisions out of all supported data formats in GPUs. Subsequently, we provide our FP-MP quantization methodology to outperform full-precision.

**1) Precision Selection:** In recent years, GPU vendors have introduced support for a variety of data formats. The supported floating-point formats include FP32, TF32, BF16, FP16, FP8 (E4M3), FP8 (E5M2), FP6, and FP4. This creates a large search space. Although the last two formats are not the focus of this work, our approach is applicable to them as well. To simplify precision selection, we limit the options to two precision levels: *low* and *high*. To determine these levels, we conduct experiments on three different diffusion models, where we quantize all linear layers to a specific data format in each case (see Fig. 2). Experimental settings are discussed in Sec. 5.1. We find that 32-bit and 16-bit quantization offer comparable performance, leading us to prefer 16-bit for its significant speedup. Between BF16 and FP16, BF16 performs better in most cases, making it our choice for *high-precision*. For *low-precision*, we select FP8 (E4M3) over FP8 (E5M2) due to better performance across more scenarios. Our key insight is that reducing exponent bits beyond a certain threshold severely degrades image generation quality due to large errors from overflow, which propagate through diffusion model timesteps. In contrast, reducing mantissa bits does not cause significant errors and can even enhance image diversity. However, for 8-bit formats, a 2-bit mantissa is insufficient to maintain precision.

**2) Search Space Reduction:** The challenge of mixed-precision (FP-MP) search stems from the vastness of the search space: even with just two formats (BF16 and FP8), $T$ timesteps, and $L$ layers, the search space expands to an overwhelming $2^{(T \times L)}$ possibilities. Given the impracticality of a brute-force approach, we further reduce the search space by categorizing the layers into two distinct sets: sensitive and insensitive. Sensitive layers are those for which quantization significantly degrades

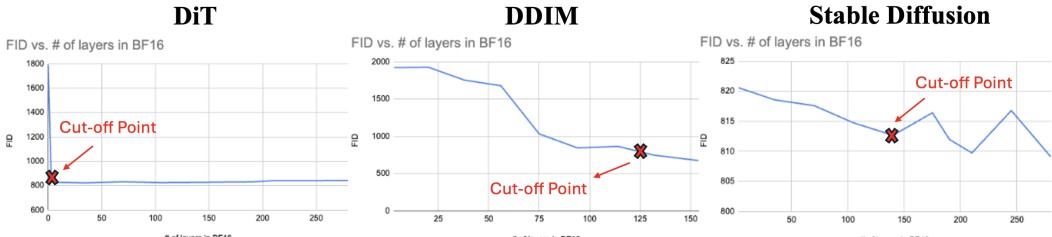

Figure 3: Selecting cut-off point between high and low precision for different models: X-axis is the number of layers quantized to high-precision (the rest of the layers are quantized to low-precision) and Y-axis is FID (lower is better).

model quality, necessitating their preservation in *high precision*. Conversely, insensitive layers can be safely quantized to a *low precision* without severely impacting the model's performance.

**3) Sensitivity Criteria:** To determine which layers fall into the sensitive category, we apply two criteria. *(1) Range*: layers exhibiting a large range are prone to substantial errors when quantized, as values may be clipped to a maximum threshold. *(2) Standard Deviation (STD) of distribution*: Layers with a high STD are susceptible to large quantization errors due to the limited resolution of low-precision formats. Using these criteria, we sort the layers by their sensitivity. This sorted list provides a prioritized guide for quantization. Algorithm 1 outlines our methodology for identifying the sensitive layers in diffusion models.

---

**Algorithm 1** DM-Tune Algorithm for Identifying Sensitive Layers

---

1: **Input:** Calibration dataset, MAX_FLT (maximum allowable value). **Output:** `sensitive_list`
2: `sensitive_list` ← [], Shuffle the calibration dataset.
3: **for** each batch of samples in the calibration dataset **do**
4:    Randomly select a seed.
5:    Calculate the running average of overflow ratios for activation layers, storing in `overflow_layers`.
6:    Calculate the running average of std for activation layers, storing in `std_layers`.
7: **end for**
8: Sort `overflow_layers` by ratio, `std_layers` by std. i ← 0
9: **while** `overflow_layers[i].ratio` ≠ 0 **do**
10:    Push `overflow_layers[i].layer` to `sensitive_list`, i ← i + 1
11: **end while**
12: **for** i = 0 **to** size(`std_layers`) - 1 **do**
13:    **if** `std_layers[i].layer` is not in `sensitive_list` **then**
14:       Push `std_layers[i].layer` to `sensitive_list`.
15:    **end if**
16: **end for**
17: **return** `sensitive_list`

---

**4) Selecting Cutoff Point:** Once the layers are sorted by sensitivity, the next step is to determine an appropriate cutoff point between *high* and *low* precision. However, evaluating each potential cutoff point by running the model and measuring quality would be prohibitively time-consuming. To address this, we employ a *binary search* technique to efficiently converge on a cutoff point where the model's quality is close to that of full-precision. We begin by selecting the midpoint of the sorted sensitivity list and evaluating the model's performance. If the quality falls below a specified threshold, we proceed to the midpoint of the upper half of the list. Otherwise, we move to the midpoint of the lower half, progressively narrowing the range. This process is repeated until the range is reduced to a single layer. Figure 3 depicts the cut-off point determined using this approach for three different models. Each model has distinct requirements regarding the number of sensitive layers. For instance, while DiT requires only a few high-precision layers to maintain quality, DDIM needs most layers in high-precision. Although this approach helps us achieve near-full-precision quality, it alone may not be sufficient to surpass it.

**5) Surpassing Full-Precision:** To outperform full-precision, we introduce two additional techniques: *(1) Prompt-aware quantization*: conditional diffusion models often employ classifier-free guidance Ho & Salimans (2022), which requires the model to process two inputs: one with the given prompt and another with a null prompt. In this technique, we propose to quantize only the path associated with the input prompt to low-precision, while maintaining the null prompt path in

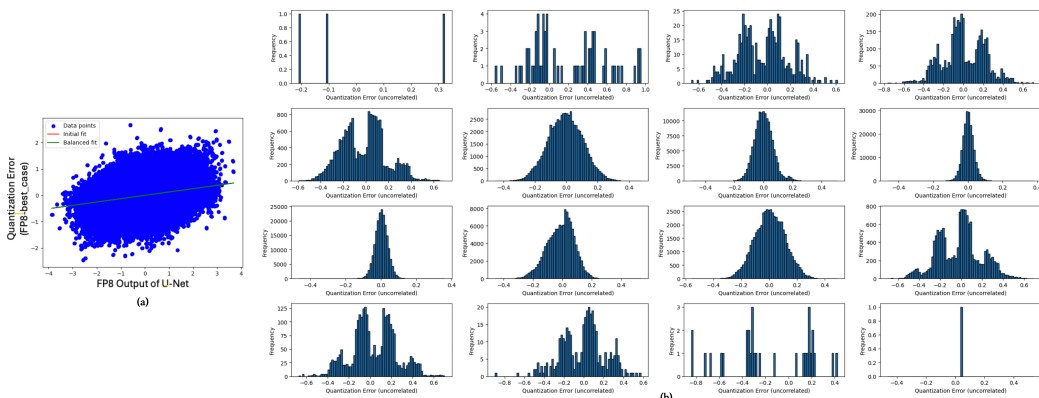

Figure 4: (a) U-Net output quantization noise at timestep=12 for Stable Diffusion model using MS-COCO dataset: it is modeled with a correlated (linear) and an uncorrelated (nonlinear) component. (b) Distribution of the uncorrelated component for different ranges of U-Net output.

high-precision. This selective quantization introduces controlled noise into the image generation process, which can enhance the overall quality of the output. *(2) Timestep-aware quantization*: this approach focuses on the quantization of insensitive layers during the early timesteps (the first 80% in this work) of the diffusion process to low-precision, while preserving them in high-precision for the final timesteps. This strategy allows the model to recover from any quantization-induced noise introduced in the early steps, ensuring sharper and more accurate images. Failure to preserve precision during the final timesteps has been shown to result in blurriness.

By following the five steps outlined in this section, we achieve our proposed FP-MP design, which not only approaches the quality of full-precision but also exceeds it. To summarize, only the portion of the activation tensors that receives the prompt during early timesteps for insensitive layers (as determined by Algorithm 1 and our cutoff point) are quantized to FP8 (E4M3), while the rest remain in BF16. For the weights, all sensitive (insensitive) layers are quantized to BF16 (FP8).

## 4.2 Noise Modeling

The goal of noise-tuning is to reconstruct FP-MP using only unified low-precision (FP8 in this work) with adjustable noise. To efficiently derive such noise and minimize the number of added parameters, we aim to establish a relationship between the U-Net low-precision output and the quantization error due to transitioning from FP-MP to unified low-precision. Previous work He et al. (2024b) demonstrated that the error is linearly correlated with the U-Net output for INT8 quantization. We conduct an experiment to profile the FP quantization error. Figure 4 (a) shows this relationship for a specific timestep of the Stable Diffusion model using MS-COCO calibration dataset. We observe that there is a *nonlinear* relationship between U-Net output in unified low-precision and quantization error, which differs from integer quantization. As shown, the error can be modeled with a correlated (linear) and an uncorrelated component (distance from the fitted line). Figure 4 (b) displays the distribution of the uncorrelated component across different ranges of U-Net output. We find that three overlapping Gasussians are needed to effectively model this uncorrelated part. Thus, we parameterize each Gaussian with trainable tensors that represent the mean, variance, and scaling of each distribution. We formulate the quantization noise as:

$$\mathbf{x}_{t-1,MP} = \frac{1}{\sqrt{\alpha_t}}\left(\mathbf{x}_{t,MP} - \frac{\beta_t}{\sqrt{1-\bar{\alpha}_t}}\left(\epsilon_{\theta,LP}(\mathbf{x}_{t,MP},t) + \Delta\epsilon_\theta(\mathbf{x}_{t,MP},t)\right)\right) + \sigma_t\mathbf{z} \quad (6)$$

$$\Delta\epsilon_\theta(\mathbf{x}_{t,MP},t) = P_{t,0}\cdot\epsilon_{\theta,LP}(\mathbf{x}_{t,MP},t) + \sum_{i=0}^2 P_{t,1+3\cdot i}\cdot\exp\left(-0.5\left(\frac{P_{t,2+3\cdot i}-\epsilon_{\theta,LP}(\mathbf{x}_{t,MP},t)}{P_{t,3+3\cdot i}}\right)^2\right) (7)$$

where $\mathbf{x}_{t,MP}$, $\mathbf{z}$, $\epsilon_{\theta,LP}$, $\Delta\epsilon_\theta$ represent data sample at timestep $t$ with FP-MP, a sample from distribution $\mathcal{N}(0,\mathbf{I})$, U-Net output in low-precision, and quantization error (low-precision and FP-MP U-Net output difference). $\alpha_t$, $\beta_t$, and $\sigma_t$ are hyperparamters. $P_{t,j} \in \mathbb{R}^{C \times H \times W}$ ($j \in \{0,1,\ldots,9\}$) is $j^{th}$ trainable parameter at timestep $t$. It has the same dimension as $\mathbf{x}_t$ with $C$, $H$, and $W$ representing the number of channels, height, and width of the latent.

### 4.3 DATA GENERATION

In this stage, we generate the necessary training data for the subsequent noise-tuning step. First, we curate the input prompts: for conditional models, we randomly sample prompts from the evaluation dataset, while for unconditional models, input prompts are not required. For conditional models, since the FP-MP design (Sec. 4.1) outperforms full-precision, we use the FP-MP U-Net output as the ground truth for training. In contrast, for unconditional models, we use the U-Net output in full-precision as the ground truth. This is because, without prompts, it is not possible to leverage prompt-aware quantization, leading to FP-MP performance being inferior to full-precision. We run the model using FP-MP (for conditional models) or full-precision (for unconditional models) and record the U-Net outputs at each time step. This procedure is essential for ensuring that the generated data provides an accurate and effective ground truth for the next phase of training.

### 4.4 NOISE-TUNING

The noise-tuning phase is a crucial component in our framework, where we aim to automate the fine-tuning of noise-adjusting parameters introduced in the U-Net head. Unlike traditional manual methods of fitting lines or curves to model behavior, our approach leverages the power of data-driven learning, guided by the ground truth data generated in Sec. 4.3. We only train the newly added parameters responsible for noise tuning, while keeping the rest of the model frozen. It minimizes the risk of overfitting and reduces the computational burden, as only a small subset of parameters is optimized. Training is conducted over a predefined number of epochs, with convergence criteria established to halt training if improvements plateau.

### 4.5 RUNTIME OPTIMIZATION

Since the nonlinear function introduced in noise-tuning is memory-bound, it hinders the overall speed of DM-Tune. With the recent quantization of the smaller number of bits, these memory-bound kernels become more pronounced. Thus, we develop an optimized GPU kernel to fuse the low-precision matrix multiplication with Gaussian terms. We implement matrix multiplication in FP8 using CUTLASS[1], leveraging high-throughput tensor cores. The computations in the noise-tuning head involve performing matrix multiplication, casting the output to full-precision, adding it to the Gaussian terms, and then casting it back to FP8. Intrinsic functions are employed for casting and calculating the exponential. We apply three optimizations to further improve the performance: using lookup tables (LUTs) to replace the whole complex function of noise-tuning (O1), data prefetching (O2), and utilizing vector instructions (O3). We describe each optimization in more detail. **O1:** We completely replace the Eq. 7 with LUTs implemented as shared memory. Based on the profiling results (i.e., Fig. 4 (a)), we set a range for the values of $\epsilon_{\theta,LP}(\mathbf{x}_{t,MP}, t)$ and allocate 32 KB shared memory. We first calculate the LUT index based on the value of $\epsilon_{\theta,LP}(\mathbf{x}_{t,MP}, t)$ and retrieve from LUT if the index is within the acceptable range of addresses. We find that the approximation error due to using LUT is negligible. **O2:** We overlap the time for calculating exponential functions with loading data from global memory. We use rolling prefetch; prefetching one element and calculating exponentials for another element. **O3:** We use vector instructions which improve instruction-level parallelism (ILP), throughput, and memory coalescing. We found that grouping eight consecutive elements in FP8 leads to the highest throughput. Once combined with O2, we prefetch eight elements at once and calculate the exponentials for the other eight elements.

## 5 EXPERIMENTAL RESULTS

### 5.1 EXPERIMENTAL SETUP

**Models and datasets:** We use three different types of diffusion models: Stable Diffusion v1.5 Rombach et al. (2022a) (text-conditioned), Diffusion Transformer (DiT) Peebles & Xie (2023) as a class-conditioned model, and DDIM Song et al. (2021) that is an unconditional model. The datasets are MS COCO Lin et al. (2015), ImageNet Deng et al. (2009), and CelebA-HQ (loaded from Hugging Face[2]), respectively. The resolution of the images in the first model is 512×512 and it is 256×256 for the other two models. All experimental configurations (e.g., variance schedule, guidance scale, and scheduler) follow the official implementation. The number of time steps is 50. We also use LDM Rombach et al. (2022a), and IDDPM Nichol & Dhariwal (2021) for comparing against related work.

---

[1]https://github.com/NVIDIA/cutlass
[2]https://huggingface.co/google/ddpm-celebahq-256

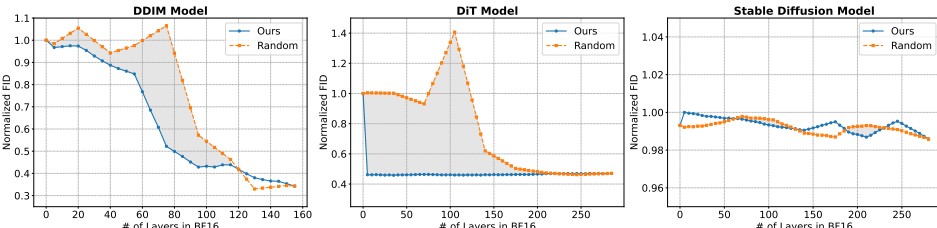

Figure 5: Comparison of FID between our sensitivity-based quantization strategy and random precision selection, gradually shifting from entirely low-precision to entirely high-precision in the layers.

**Quantization baselines:** We compare the performance of our approach against these state-of-the-art: Q-Diffusion Li et al. (2023a), PTQ4DM Shang et al. (2023), and PTQD He et al. (2024b).

**Implementation:** We implement DM-Tune using PyTorch. To develop a model-agnostic quantization method, we employ hook functions, registering them conditionally only to specific layers, timesteps, and portions of tensors targeted for quantization. During forward pass, activation tensors are modified by performing quantization followed by de-quantization. During the backward pass, straight-through estimation (STE) Yin et al. (2019) is performed. Batch normalizations and activation functions are kept in high-precision. We quantize both activations and weights to the desired format. For runtime evaluation, we developed highly optimized low-precision kernels with support for noise-tuning, building on the original CUTLASS repository. For noise-tuning and evaluating the quality of image generation, we run the models on an A100 GPU (40 GB memory) as it provides enough memory. For assessing runtime, we run the models on an L4 GPU (24 GB memory) because it supports FP8 precision. We evaluate the diffusion models based on DGM-Eval Stein et al. (2023). We keep the seed fixed when comparing different quantization methods. For noise-tuning, Adam optimizer is used with a learning rate of *1e-3* with 4K samples for training and 1K samples for evaluation. The batch sizes for evaluating diffusion models are 8 for Stable Diffusion, 64 for DiT, and 32 for DDIM. However, for noise-tuning, we halved them to prevent out-of-memory (OoM) errors.

## 5.2 MIXED-PRECISION SENSITIVITY

We first demonstrate the effectiveness of our mixed-precision sensitivity analysis before applying prompt-aware and timestep-aware quantization techniques. We evaluate the sensitivity criteria by comparing it to a random method, where we randomly decide which layers should be in high-precision and which in low-precision based on a predetermined split. Figure 5 compares the FID between our sensitivity-based approach and random strategy for three models. For DiT, our approach quickly identifies sensitive layers, resulting in a larger gap between our method and the random approach. The gap is smaller for DDIM. For Stable Diffusion, however, the choice of layers for low- and high-precision quantization has minimal impact on final accuracy.

## 5.3 TRAINING

We evaluate noise-tuning performance over training epochs and compare it with the best mixed-precision design (FP-MP ground truth), FP8, and FP32, to assess its convergence. Additionally, to emphasize the superior expressivity of our noise-tuning approach, we compare it to a version of noise-tuning that uses only one Gaussian term instead of three. Figure 6 shows the training performance (FID metric) across epochs for different approaches using the MS COCO evaluation dataset in the Stable Diffusion model. "*NT 1_Gaussian*" and "*NT 3_Gaussian*" are noise-tuning with one and three Guassian term(s), respectively. The training setup is discussed in Sec. 5.1. As shown, noise-tuning with three Gaussians converges much faster than with one Gaussian due to its greater expressivity. Moreover, noise-tuning can even outperform the FP-MP ground truth. The results

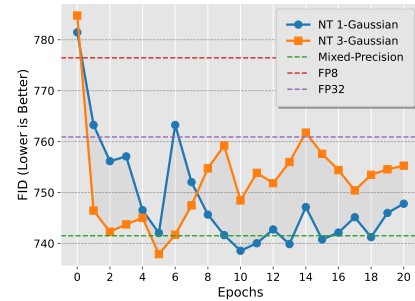

Figure 6: Comparison of FID scores across epochs for two noise-tuning methods with one and three Gaussian terms for Stable Diffusion model using MS COCO evaluation dataset.

Table 1: Comparing DM-Tune performance with different formats for Stable Diffusion, DiT, and DDIM models. NT is noise-tuning.

| Model/Dataset | Method | FID ↓ | Precision ↑ | Recall ↑ | Authenticity ↑ | CLIP ↑ | Density ↑ | Coverage ↑ | KD ↓ | SW-approx ↓ |
|---|---|---|---|---|---|---|---|---|---|---|
| Stable Diffusion (MS COCO 512×512) | FP32 (Baseline) | 760.9 | 0.92 | 0.87 | 62.79 | 31.44 | 0.91 | 0.86 | 0.077 | 0.15 |
| | FP8 | 776.44 | 0.92 | 0.86 | 67.18 | 31.56 | 0.91 | 0.88 | 0.091 | 0.16 |
| | INT8 | 3993.55 | 0.0 | 0.0 | 100.0 | 19.16 | 0.0 | 0.0 | 27.62 | 1.73 |
| | Q-Diffusion (W8A8) | 757.51 | 0.92 | 0.87 | 63.14 | 31.40 | 0.88 | 0.86 | 0.077 | 0.15 |
| | Q-Diffusion (W4A8) | 768.90 | 0.92 | 0.83 | 63.57 | 31.27 | 0.92 | 0.87 | 0.085 | 0.16 |
| | Ours (w/o NT) | 741.51 | **0.94** | 0.86 | 60.94 | **31.58** | **0.95** | **0.89** | **0.070** | **0.14** |
| | Ours (w/ NT) | **737.89** | 0.91 | **0.89** | 64.31 | 31.32 | 0.94 | 0.92 | 0.075 | 0.14 |
| DiT (ImageNet 256×256) | FP32 (Baseline) | 721.58 | 0.99 | 0.97 | 30.78 | 30.10 | 0.89 | 0.98 | 0.045 | 0.14 |
| | FP8 | 1626.56 | 0.98 | 0.31 | 70.41 | 24.31 | 0.75 | 0.91 | 1.88 | 0.63 |
| | INT8 | 4300.40 | 0.0 | 0.0 | 100.0 | 19.16 | 0.0 | 0.0 | 32.1 | 1.91 |
| | Ours (w/o NT) | 704.74 | 0.99 | 0.97 | **33.77** | 29.91 | **0.94** | **0.99** | 0.045 | **0.13** |
| | Ours (w/ NT) | **701.23** | 0.98 | 0.97 | 32.94 | **30.87** | 0.99 | 0.96 | **0.041** | **0.12** |
| DDIM (CelebAHQ 256×256) | FP32 (Baseline) | 659.35 | 0.85 | 0.39 | 89.55 | N/A | 0.71 | 0.55 | 2.16 | 0.48 |
| | FP8 | 1971.28 | 0.28 | 0.02 | 98.24 | N/A | 0.08 | 0.05 | 8.16 | 1.02 |
| | INT8 | 4379.36 | 0.0 | 0.0 | 100.0 | N/A | 0.0 | 0.0 | 40.0 | 2.07 |
| | Ours (w/o NT) | 750.16 | 0.80 | 0.29 | **91.02** | N/A | 0.54 | 0.45 | 2.41 | 0.51 |
| | Ours (w/ NT) | **653.98** | 0.82 | **0.44** | 90.14 | N/A | 0.66 | **0.58** | 2.18 | **0.45** |

Table 2: Comparing DM-Tune performance with state-of-the-art. NT is noise-tuning.

| Model/Dataset | Method | FID ↓ | Precision ↑ | Recall ↑ | Density ↑ | Coverage ↑ | KD ↓ | SW-approx ↓ |
|---|---|---|---|---|---|---|---|---|
| IDDPM (ImageNet 64×64) | FP32 (Baseline) | 282.02 | 0.80 | 0.84 | 0.77 | 0.81 | 0.15 | 0.20 |
| | FP8 | 519.50 | 0.73 | 0.56 | 0.68 | 0.48 | 0.66 | 0.36 |
| | INT8 | 4186.98 | 0.0 | 0.0 | 0.0 | 0.0 | 31.23 | 1.84 |
| | PTQ4DM | 376.09 | 0.78 | 0.80 | 0.78 | 0.72 | 0.29 | 0.27 |
| | Ours (w/o NT) | **256.17** | **0.82** | **0.87** | 0.74 | **0.84** | **0.14** | **0.16** |
| LDM (ImageNet 64×64) | FP32 (Baseline) | 265.83 | 0.98 | 0.42 | 0.98 | 0.89 | 0.059 | 0.12 |
| | FP8 | 346.77 | 0.95 | 0.41 | 0.99 | 0.87 | 0.28 | 0.26 |
| | INT8 | 3519.99 | 0.45 | 0.0 | 0.11 | 0.0 | 19.35 | 1.52 |
| | PTQD | 226.51 | 0.98 | 0.53 | 0.97 | 0.91 | 0.041 | 0.10 |
| | Ours (w/o NT) | **232.79** | 0.96 | **0.51** | 0.98 | **0.93** | **0.048** | **0.09** |

suggest that the FP-MP performance can be achieved in just 5 to 10 epochs with noise-tuning, highlighting the efficiency and low training cost of our approach.

## 5.4 IMAGE GENERATION PERFORMANCE

In this subsection, we comprehensively assess the performance of DM-Tune and compare it with other quantization methods. Table 1 shows this performance comparison for three diffusion models using 1K samples based on DGM-Eval evaluation. For both Stable Diffusion and DiT models, our approach without noise-tuning (NT) outperforms full-precision, and noise-tuning typically further enhances performance. However, for DDIM, which is an unconditional model, DM-Tune without noise-tuning does not surpass the FP32 baseline. With noise-tuning, though, it provides a performance boost for certain evaluation metrics. We presume that this performance advantage arises because we initially introduce some noise through mixed-precision (prior to noise-tuning), and by subsequently adding nonlinear parameters and fine-tuning the model, we achieve improved results. Overall, DM-Tune outperforms full-precision in terms of general quality, diversity, prompt alignment, and originality. Comparing conditional and unconditional diffusion models, our approach shows greater effectiveness for conditional models, where we progressively apply controlled noise to the prompt. In contrast, the improvement for unconditional models is comparatively marginal. In addition, our approach provides better performance compared to Q-Diffusion for most cases. This is partly due to the fact that it is an integer-based quantization method with a limited dynamic range, and it does not employ techniques to surpass full-precision image quality. As shown in Figure 1 (a), I-MP struggles to generate high-fidelity images, particularly when it comes to human faces.

Since we use DINOv2-ViT model as the feature extractor to avoid unfairness and bias in the evaluation, our FID results are larger than those published in related work. Thus, we also regenerate the related work performance with our evaluation approach using the same feature extractor. However, since related work only support specific models and datasets due to their calibration method, we separately compare DM-Tune with them for the datasets they support. Table 2 shows this comparison using 5K samples based on DGM-Eval evaluation. As shown, DM-Tune outperforms prior arts for most of the scenarios. Among SOTA, only PTQD provides better performance compared to our approach for some metrics.

## 5.5 RUNTIME

We now assess the runtime of the proposed noise-tuning. First, we focus on the runtime of a single matrix-multiplication kernel with NT head. Table 3 presents the performance comparison of various optimizations applied to a matrix multiplication kernel fused with noise-tuning (with three Gaussian terms) where the dimensions of all input and output matrices are 1k×1k.

Table 3: Comparison of noise-tuning optimization techniques for matrix multiplication (1k×1k matrices). Normalized runtime is shown.

| FP8 | NT (8 bit) w/o opt | NT (8 bit) O1 | NT (8 bit) O1 + O2 | NT (8 bit) O1 + O2 + O3 |
|---|---|---|---|---|
| 1.00X | 1.12X | 1.04X | 1.02X | 1.01X |

We run each kernel 10 times and then report the average runtime compared to FP8 (with no noise-tuning). By enabling all of the optimizations, the runtime becomes similar to that of FP8 with no noise-tuning, which demonstrates the effectiveness of our optimizations in mitigating the performance loss associated with noise-tuning.

Next, we compare the model performance of our optimized noise-tuning with FP32, FP8, and the state-of-the-art as shown in Figure 7. The average per-sample runtime is measured and normalized to that of FP32. The proposed fully optimized noise-tuning introduces minimal overhead compared to FP8, while boosting performance by an average of $5.2\times$ compared to prior arts in diffusion model quantization.

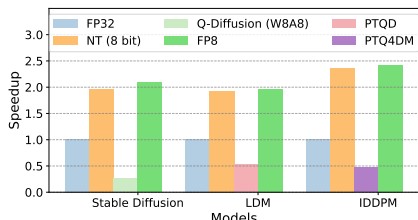

Figure 7: Normalized model performance comparison of noise-tuning against state-of-the-art methods.

## 6 RELATED WORK

**Diffusion Model Quantization:** Q-Diffusion Li et al. (2023a) proposes a post-training quantization (PTQ) strategy for integer format (8/4 bits) with a time step-aware calibration data sampling mechanism from the pretrained diffusion model. Similarly, PTQ4DM Shang et al. (2023) also employs 8-bit integer quantization for diffusion models, but its focus is limited to smaller models like DDPM and DDIM. To the best of our knowledge, no prior work has shown quantization to be beneficial for the image generation process in diffusion models.

**Fine-Tuning:** QuEST Wang et al. (2024) proposes a fine-tuning framework for low-bit diffusion model quantization to enhance model robustness against large activation perturbations under the supervision of the full-precision. The authors in Li et al. (2023b) develop a timestep-aware smoothing process to avoid oscillations in the activation distribution that can work for both inference and training. Again only small-scale models are considered. The disadvantage is that it is more resource-intensive than training a full-precision model. Efficient-DM He et al. (2024a) further uses low-rank adapters (LoRA) Hu et al. (2021) to reduce training costs. However, it introduces extra weight parameters and still requires substantial training iterations.

**Noise Modeling:** The authors in He et al. (2024b) propose a mixed-precision quantization method based on integer format that models the quantization noise and utilizes correction methods to reconstruct the full-precision output. However, they assume that the full-precision provides the highest performance as opposed to our approach which proposes to use mixed-precision as ground truth. Also, their approach is based on statistics and manual profiling which limits their applicability to certain data formats and models.

## 7 CONCLUSION

In this work, we proposed DM-Tune, a novel framework that combines unified low-precision quantization with noise-tuning to enhance the efficiency of diffusion models while improving image quality. Our approach overcomes the limitations of traditional mixed-precision quantization by effectively utilizing the inherent noise in diffusion models to improve image generation metrics. Through extensive experiments on multiple diffusion models, we demonstrated that DM-Tune not only matches but can outperform full-precision models in terms of image quality, diversity, and text-to-image alignment, while significantly reducing inference time. The optimized GPU kernels further accelerate deployment of diffusion models. Future work will explore extending DM-Tune to even lower-precision formats such as FP4/6 and quantizing additional components of diffusion models, such as the encoder-decoder structures, to further enhance runtime.

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
