# OpenReview forum: "DM-Tune: Quantizing Diffusion Models with Mixture-of-Gaussian Guided Noise Tuning"
_ICLR.cc/2025/Conference — ICLR 2025 Conference Withdrawn Submission_

### Official Review · Reviewer_WAAM · 2024-10-30

**Soundness:** 3
**Presentation:** 2
**Contribution:** 3
**Rating:** 5
**Confidence:** 3

**Summary:**

The paper presents a novel framework to improve the efficiency of diffusion models by combining unified low-precision quantization with a noise-tuning mechanism. DM-Tune optimizes image generation by using trainable Gaussian noise adjustments to recover the quality lost from low-precision outputs, achieving results that are close to both full-precision models and existing mixed-precision methods. It introduces a simple-yet-effective approach to detect sensitive layers together with prompt-aware and timestep-aware quantization to optimize selected layers within the iterative diffusion process for precision adjustments, ensuring sharper outputs. Additionally, the paper develops optimized GPU kernels to enhance runtime performance, significantly reducing computational overhead. Experimental results show DM-Tune improves image quality, diversity, and text-to-image alignment while speeding up inference, making it a practical solution for deploying diffusion models efficiently.

**Strengths:**

The paper offers a unique approach by integrating unified low-precision quantization with Gaussian-guided noise-tuning, an innovative mechanism that allows low-precision outputs to even surpass full-precision performance.

The research is thorough, with well-structured experiments demonstrating improvements in both image quality and runtime performance. The use of multiple diffusion models (Stable Diffusion, DiT, and DDIM) and datasets enhances the robustness of the findings.

The paper is generally well-organized, with clear descriptions of key components such as noise-tuning, sensitivity-based quantization, and runtime optimization.

The paper addresses a critical challenge in deploying large-scale diffusion models, making it highly relevant for real-world applications requiring efficient, high-quality generative outputs. With growing interest in resource-efficient AI, DM-Tune’s ability to improve performance while reducing computational costs makes it a significant contribution.

**Weaknesses:**

1. While the unified low-precision quantization strategy is effective, the scalability of the approach—especially when applied to larger diffusion models beyond those tested—is unclear. Including runtime and memory trade-offs when scaling to more complex models (e.g., SDXL) or higher-resolution tasks would enhance the practical utility of the work.

2. There is a lack of BF16 (baseline) when the authors try to demonstrate the effectiveness of the purposed method on FP8 configurations.

3. In Fig.5,  as shown in the third figure, the proposed sensitive-layer selection against randomized selection does not make too much difference in terms of StableDiffusion and the authors do not further discuss such an observation. Besides, there is a lack of mathematical or theoretical justification for the proposed Algorithm.1.

**Questions:**

According to the weakness section, I have the following questions:

1. How does the proposed DM-Tune framework perform on larger diffusion models, such as SDXL, or tasks requiring higher resolution? Can the authors provide a detailed analysis of the runtime and memory trade-offs when scaling to these more complex scenarios to validate the scalability of the approach?

2. Why is BF16 omitted as a baseline together with FP32 baseline when evaluating the effectiveness of FP8 configurations?

3. why does the proposed sensitive-layer selection strategy show minimal improvement over randomized selection for the Stable Diffusion model? What insights do the authors have regarding this outcome?

4. Some of the attached figures are really tough to follow. For Fig.1 & Fig.2, even though I zoomed in for the max extend, I still cannot get a clear visualization of the generated images. For Fig.2, the lines are too close to be distinguished.

5. The pseudocode in Alogrithm.1 depicts the fact that overflow detection is the first criteria when it comes to sensitive layer selection. But how STD actually affects the sensitivity of the certain layer is remain unclear in the paper.

6. The paper could be more demonstrating if real world testing prompts could be attached for visualization comparison between the FP8(proposed low-precision) and baselines.

---

### Official Review · Reviewer_a8Gb · 2024-10-31

**Soundness:** 2
**Presentation:** 2
**Contribution:** 3
**Rating:** 3
**Confidence:** 4

**Summary:**

This paper proposes a framework for improving diffusion model inference speed without sacrificing image quality via a tuned mixed-precision strategy. Their approach follows a study of layer sensitivity to high (BF16) and low (FP8) precision formats, across various pre-trained models and sampling noise levels (timesteps). The authors further improve image quality with a learned Gaussian approximation, and computational efficiency with custom CUDA kernels, achieving a $\sim 2\times$ inference speedup over FP32.

While this paper proposes an interesting solution to diffusion model quantization, the specific implementation details are ambiguous, and the chosen evaluation methodology may be subject to high statistical noise. Additionally, the authors make a bold claim that lower precision performs better, which should be backed up by both theoretical and empirical evidence (other than final evaluation), which would greatly increase the paper’s impact.

Overall, this paper falls below the expected threshold; however, I would be inclined to raise my score if most of the weak points are addressed upon revision.

**Strengths:**

The authors propose several interesting ideas, namely: demonstrating the robust behavior of DiT with low-precision arithmetic, insight into the use of e4m3 vs e5m2, using different precisions for conditional and unconditional paths, and approximating quantization error with a series of Gaussians. These results are backed by the use of DGM (DINOv2 from Stein et al., 2023), which will provide lower sensitivity to imperceptible image artifacts. Additionally, the authors demonstrate a significant inference speedup with their optimized CUDA kernels.

**Weaknesses:**

-	The theoretical motivation (Eqn 5,6, and 7) lacks a clear mapping to the proposed noise modeling strategy, including contradiction with the described kernels. Is the Gaussian series applied per layer, or once to the entire model? It is also unclear whether the quantization error can be accurately modeled by a Gaussian series.
-	The model performance evaluation is conducted with a low number of image samples (1k), which raises questions about statistical errors in the evaluation process. These errors may affect the conclusions drawn, potentially resulting in statistically insignificant performance improvements. Furthermore, the authors fail to reproduce the metrics from Stein et al. under the FP32 case (likely due to the low sample number).
-	The paper lacks details on the implementation, and clear analysis of findings from the main text. These details should be included in an appendix, such as pseudocode for the NT layers, pseudocode for the kernel optimizations, which layers were the most sensitive to precision change, and why the mixed-precision method performs better than full precision.
-	The kernel performance evaluation in Section 5.5 lacks rigor, where 10 calls is insufficient to establish a mean execution time.
-	The claims regarding integer quantization over floating-point are too strong. While It may be more challenging to find a good quantization scheme with uniform spacing, this does not imply that the non-linear spacing from floating-point will always perform better. Stating that the simple implementation (equivalent in complexity to FP8) does not perform as well would be sufficient, but should not be used to support a more general statement.

**Questions:**

Major Questions:

See weaknesses.

Additional Questions:

1)	There are visible artifacts in your FP-MP samples in Figure 1 (e.g. the robot is less cohesive). While it clearly performs better than I-MP, it is not immediately clear that the evaluation metrics are true representations of the generated images. Could you address this discrepancy?
2)	Your theoretical noise modeling applies directly to $X_t$ rather than the internal layers. The two are only equivalent if the model is linear, which is not the case. Could you address this or explain why the same approximation applies to the hidden activations?
3)	Why did you keep the unconditional path at full precision instead of the conditional path? It seems like the inverse may perform better, and empirical evidence supporting either choice would strengthen your argument.
4)	Does splitting the quantization per path negatively impact computational performance? It will reduce parallelism for the quantized layers.
5)	Could you provide an explanation as to why you use the standard deviation to measure the sensitivity of precision? It’s not clear to me that a wider distribution indicates increased sensitivity when fixed layer-wise scaling could be applied. Other formats such as NF4 (Dettmers et al., 2023) take advantage of this fact for non-uniform spacing.
6)	You show that DiT performs exceptionally well with quantization; however, is the same true for text-conditioned DiTs (e.g., Pixart-alpha, SD3, etc.)? If it is true, then it would be valuable to add this claim, otherwise if the behavior is not identical, consider adding a text conditioned DiT along with the other 3 models.
7)	In Figure 6, your results suggest equivalent performance with NT-1 and NT-3, although NT-1 requires more epochs to converge. Is there any real advantage to NT-3 if it necessarily has a higher computational overhead during inference? The number of tuning epochs seems largely irrelevant at this scale, as it is presumably done once per model.
8)	Were the evaluation results in Tables 1 and 2 generated with CFG? If not, please do so. Model behavior can vary drastically with and without CFG. Including both with and without would be preferable, where simplifying the columns could save space while presenting the full table in the appendix. Furthermore, please state the sampler used since that too can have a significant impact.


Comments for Clarity:


-	The motivation sentence in Section 3.1 discussing diffusion models as a stochastic process is unclear. Consider rewriting.
- In Section 4.1.1, you mention "quantization is applied to all linear layers", please clarify if this includes convolutions for the conv-nets.
-	In Section 4.5, please explain what is meant by “O3: we use vector instructions”, since warps always execute at the vector level (SIMT). Consider rephrasing.
-	Mentioning batch size for evaluation in Section 5.1 is irrelevant, where you should instead directly state the batch sizes used for tuning.
-	The $5.2\times$ speedup in Section 5.5 should be clarified by directly stating the reference point. Additionally, consider using an arrow indication in Figure 7 for easy visual clarity.

Comments for Presentation:

-	In Figure 1a, which method is yours? This should be clearly marked and included if not done so.
-	Non-cherry-picked samples should be added to the appendix to better compare the different versions (I-MP can be omitted in this case as your focus is on FP-MP).
-	Figure 2 is difficult to read as the curves often overlap, applying a relative scaling may improve legibility.
-	The critical point in Figure 3a is not legible due to the axis scale, consider using log scale or adding a sub-frame-region around the origin.
-	Figure 7 should use a pattern fills along with color coding to improve readability. Also consider adding the geomean, which is typical to aggregate overall performance.
-	Ensure consistency in the ordering of figures. Every time you show DiT, SD, and DDIM, they should appear in the same order from left to right.
-	The plot titles in Figure 3 (“FID vs…”) should be removed. Additionally, both axis and label font sizes are too small in most figures.

---

### Official Review · Reviewer_pTx7 · 2024-11-02

**Soundness:** 2
**Presentation:** 1
**Contribution:** 2
**Rating:** 3
**Confidence:** 5

**Summary:**

This paper proposes DM-Tune, a novel approach that replaces mixed-precision quantization with a unified low-precision format through noise-tuning. The proposed method first builds a precise mixed-precision quantized model that achieves slightly better image generation quality than full-precision models by identifying sensitive layers. Following this, a noise-tuning scheme is applied to enable fully low-precision quantization. The noise-tuning is achieved through trainable Gaussian models. As a result, the proposed method successfully yields fully FP8 models with image generation quality comparable to full-precision models. Additionally, the authors have developed a custom CUDA kernel to mitigate the overhead of the noise-tuning scheme during image generation.

**Strengths:**

1.	This paper effectively identifies a mixed-precision model configuration that preserves image generation quality.
2.	The proposed noise-tuning scheme successfully calibrates quantization noise by leveraging the noise estimation stage of diffusion models.
3.	To minimize the overhead introduced by noise-tuning, custom CUDA kernels are developed to fuse noise-tuning operations with matrix multiplication.

**Weaknesses:**

1.	The runtime and size of the calibration dataset required to identify the optimal mixed-precision model are not discussed.
2.	Details regarding the training of noise-tuning parameters are lacking. Specifically, the timesteps used for training, the dataset size, memory requirements, and training time are not provided.
3.	While this paper introduces a quantization training process, it only compares the proposed method with basic PTQ approaches. Although the training procedure of the proposed method is relatively simple, recent quantization-aware training (QAT) methods for diffusion models have introduced training overheads similar to those of PTQ calibration procedures. Therefore, a comprehensive comparison with recent QAT methods, such as EfficientDM [1], is necessary. EfficientDM achieves slightly better image generation quality than full-precision models with INT8 quantization of both weights and activations and even achieves comparable image generation quality to full-precision models when further lowering weights to INT4. EfficientDM requires approximately 3 hours to train models for generating ImageNet 256x256 images using a single RTX3090 GPU.
4.	Although the concept of timestep-aware quantization has been proposed in previous work [2], this paper does not adequately reference it.
5.	NVIDIA TensorRT supports INT8 quantization based on Q-Diffusion, and INT8 quantization has been shown to be nearly twice as fast as FP16 [3]. However, in Figure 7, this paper suggests that Q-Diffusion with INT8 quantization is significantly slower than FP32. Without a proper explanation, this discrepancy in speedup results is difficult to accept.

[1] “EfficientDM: Efficient Quantization-Aware Fine-Tuning of Low-Bit Diffusion Models”, ICLR 2024.

[2] “Leveraging Early-Stage Robustness in Diffusion Models for Efficient and High-Quality Image Synthesis”, NeurIPS 2023.

[3] https://developer.nvidia.com/blog/tensorrt-accelerates-stable-diffusion-nearly-2x-faster-with-8-bit-post-training-quantization/

**Questions:**

1.	What is the total number and memory size of the trainable noise-tuning parameters that are assigned in a layerwise and timestep-wise manner?
2.	What are the overflow ratios in Algorithm 1, and how is "overflow" defined? Does it mean the overflow that occurs when converting high-precision FP to FP8?

---

### Official Review · Reviewer_tQAP · 2024-11-04

**Soundness:** 3
**Presentation:** 2
**Contribution:** 2
**Rating:** 6
**Confidence:** 3

**Summary:**

The paper aims to accelerate the execution of diffusion models through quantisation. The paper makes the argument that mixed-precision can provide better quality results than floating point full precision in the case of diffusion models, and proceeds with the introduction of a strategy in defining such MP bases on two precision levels. Driven by the overheads that occur when models of multiple precisions are computed, the authors provide a method for a unified precision. Towards that, the authors make the argument that higher precision can be generated through lower precision by adding noise in a specific way (which is learned during a training process). Finally, the authors provide an optimised GPU kernel implementation for the above step.

**Strengths:**

Interesting results in seeing that lower precision (still floating point) can lead to better performance (or at least as good as) full-precision for this specific class of problem.

The methodology is appropriate and it is clearly laid out .

**Weaknesses:**

The paper is highly experimental and there is no theoretical support of the above arguments. As such generalising these results is not obvious. Nevertheless, the authors have demonstrated the efficacy of their approach to a good number of scenarios.

Section 4.4 is on of the most crucial parts of the method, but not enough details are provided. The training approach is not well described and the paper only provides some high level information.

In eq 7, it is not clear why only 3 gaussian components are used. Is this decision based on further experiments? It would be nice to see the speed up of the system due to 4.5 (CUDA kernel).

**Questions:**

Q1: In Table 7, it is not clear why Q-Diffusion and PTQD achieve worst performance than FP32. I am assuming that FP32 denotes the original non-quantised model.

---

### Note · Authors · 2024-11-15

**Comment:**

We sincerely thank the reviewers' comments and suggestions. We will improve our work and resubmit.

**Withdrawal Confirmation:**

I have read and agree with the venue's withdrawal policy on behalf of myself and my co-authors.